# COVID 19 Fear Impact among Russian "Helping Profession" Students

**Vsevolod Konstantinov** [1,*], **Valentina Gritsenko** [2], **Raushaniia Zinurova** [3], **Elena Kulesh** [4], **Irina Osipenko** [5], **Alexander Reznik** [6] **and Richard Isralowitz** [6]

1   Department of General Psychology, Penza State University, 440026 Penza, Russia
2   Department of Social Psychology, Moscow State University of Psychology and Education, 125009 Moscow, Russia; so@so.mgppu.ru
3   Department of Management and Entrepreneurship, Kazan National Research Technological University, 420015 Kazan, Russia; rushazi@rambler.ru
4   Department of Psychology, Pacific State University, 680035 Khabarovsk, Russia; resurssentr@mail.ru
5   Department of Clinical Psychology, Smolensk State Medical University, 214019 Smolensk, Russia; osipenkoir@mail.ru
6   Regional Alcohol and Drug Abuse Research Center, Ben Gurion University of the Negev, Beer Sheva 8410501, Israel; reznikal@bgu.ac.il (A.R.); richard@bgu.ac.il (R.I.)
*   Correspondence: konstantinov_vse@mail.ru

**Abstract:** Background: Little is known about the COVID-19 impact on Russian medical, psychology and social work students' psycho-emotional well-being, substance use and resilience. Methods: More than 2000 helping profession students, 75.4% female, participated in an online survey about COVID-19 impact at a peak time of infection (October/November 2020). The Fear of COVID-19 Scale (FCV-19S) and Brief Resilience Scale (BRS) were used for study purposes. Furthermore, the influence of COVID-19 on student psycho-emotional well-being and substance use (i.e., tobacco and alcohol) was examined. Results: Medical, female and religious students reported higher fear values. Social work students reported more current substance use, including binge drinking (five or more drinks on one particular occasion). Students who reported COVID-19 associated with their psycho-emotional well-being had higher fear values. Regarding resilience, no association was found based on the student study area. However, male and non-religious students reported more resilience. Students who reported substance use and psycho-emotional problems had lower resilience values. Conclusion: COVID-19 fear and substance use differs among Russian students based on background characteristics. including gender, religiosity and study area. The FCV-19S and the BRS were found to be reliable instruments for research of COVID-19-related psycho-emotional problems, substance use and resilience. Study findings have implications for "front line" helping profession students in terms of education, training and intervention, in support of promoting their ability to address difficult conditions resulting from the pandemic and other disaster conditions in the future.

**Keywords:** Russian students; COVID-19 fear; resilience; substance use; well-being

## 1. Introduction

COVID-19 conditions have heightened interest in personality factors that may serve as a bulwark against fear, stress and other health and behavior conditions affected by the pandemic. For example, psychological resilience or the ability to psychologically or emotionally cope with a crisis or quickly return to a pre-crisis state, is increasingly seen as a protective factor [1–3]. Among university students, the harmful pandemic effects on psychological and emotional well-being, as well as resilience to cope under difficult emergency conditions have been observed [4–6].

This study aims to examine COVID-19-related fear and its association with psycho-emotional conditions, substance use as well as resilience among Russian students in the

helping professions of medicine, psychology and social work. Such students were surveyed at an infection peak occurring in October/November, 2020. It was hypothesized that COVID-19 fear among Russian students was associated with their health and wellbeing. If so, then certain personal background factors, such as gender, culturally defined religiosity and study discipline, would be linked to student mental health, substance use and resilience.

This research is the first to be conducted on Russian "helping profession" students and the impact of COVID-19 on their health and wellbeing. Furthermore, this study provides useful information for possible education, training and intervention of helping profession students to promote their resilience and ability to address the negative impacts of disaster, emergency conditions such as COVID-19.

## 2. Methods

### 2.1. Design, Participants and Procedures

The Qualtrics software platform was used for this online survey. Two scales were used for data collection purposes. The first was the seven-item Fear of COVID-19 Scale (FCV-19S) [7,8]. The levels of agreement with FCV-19S statements were evaluated by a 5-point Likert scale from 1 (strongly disagree) to 5 (strongly agree). Higher total scores correspond with more COVID-19 fear. Two questions were added to the scale to determine COVID-19's impact on university students' studies, social life and family relations ("I fear my university studies will be negatively affected by COVID-19" and "I am experiencing excess stress and anxiety due to the impact of COVID-19 on my social and family life").

The second scale used was the six-item Brief Resilience Scale (BRS) [9] to determine the level of psychological resilience among Russian students. The BRS items assess respondents' ability to bounce back or recover from stress. The levels of agreement with BRS statements were evaluated by a 5-point Likert scale. Higher total scores correspond with a higher level of resilience. The Cronbach's reliability of the FCV-19S with two additional questions was 0.848, and it was 0.874 for the BRS.

The influence of COVID-19 on student psycho-emotional well-being (i.e., depression, exhaustion, loneliness, nervousness and anger) as well as substance use, including tobacco and alcohol, was examined. In the first case, a phenomenological approach was used, based on students' subjective self-assessment of how COVID-19 affected their psycho-emotional well-being. In the second case, a short version of the Substance Use Survey Instrument (SUSI) was used [10].

The survey instruments, prepared in English, were translated to Russian and back translated. The translation method used was consistent with that described by the World Health Organization for research purposes [11]. The Russian version of the instruments appearing on the Qualtrics platform are available on request from the corresponding author of this article.

To ensure the methods proposed for this research were ethical, the Russian investigators received approval from the ethics committees of the universities involved. These ethics approval processes are equivalent to established regulations to help protect the rights and welfare of human research subjects [12]. No external grant funding was received for the study.

The universities involved in this study conduct distance learning to link students and lecturers. This capacity was used for data collection. Students were informed about the online survey and its aim. For additional explanation and other forms of support, students could contact the curators of their groups. The role of a curator in Russian universities is similar to that of a tutor in a number of Western universities. The curator accompanies a group of students assigned to him throughout their years of education at the university and helps them in solving many issues of student life. Before responding to the questions, students were assured that their responses were confidential, the survey was compliant with all ethical standards and their responses constituted consent to participate in the survey.

## 2.2. Statistical Analysis

For this study, all statistical analyses were conducted using SPSS, version 25, and JASP, a free and open-source program for statistical analysis supported by the University of Amsterdam [13]. The Pearson's chi-squared test for dichotomous variables, a t-test, and a one- and two-way ANOVA for continuous variables were used.

## 2.3. Participants

A total of 2095 students, including medical ($n = 1287$), psychology ($n = 552$) and social work ($n = 256$) students, 24.6% male and 75.4% female, completed the online questionnaire used during COVID-19 infection in October/November 2020. Table 1 provides background characteristics of the survey respondents.

**Table 1.** Demographic data.

| | Study Area | | | |
|---|---|---|---|---|
| | **Medicine** **($n = 1287$)** | **Psychology** **($n = 552$)** | **Social work** **($n = 256$)** | **Total** **($n = 2095$)** |
| Gender, % ($n$) | *** | *** | *** | |
| Male | 28.9 (371) | 14.2 (78) | 25.0 (64) | 24.6 (513) |
| Female | 71.1 (912) | 85.8 (471) | 75.0 (192) | 75.4 (1575) |
| Age, | *** | *** | *** | |
| Mean (SD) | 19.0 (2.4) | 22.9 (7.4) | 20.0 (2.0) | 19.9 (4.2) |
| Median | 19.0 | 20.0 | 20.0 | 19.0 |
| (Range) | (17–62) | (17–62) | (17–34) | (17–62) |
| Religiosity, % ($n$) | *** | *** | *** | |
| Secular | 28.6 (367) | 44.9 (248) | 43.0 (110) | 34.7 (725) |
| Non-secular | 71.4 (916) | 55.1 (304) | 57.0 (148) | 65.3 (1366) |
| Year of study, % ($n$) | *** | *** | *** | |
| First | 30.9 (398) | 18.7 (103) | 29.6 (75) | 27.6 (576) |
| Second | 58.3 (748) | 34.8 (192) | 23.3 (59) | 47.8 (999) |
| Third | 1.5 (19) | 19.4 (107) | 28.1 (71) | 9.4 (197) |
| Fourth and more | 9.3 (119) | 27.1 (150) | 19.0 (48) | 15.2 (317) |

*** $p < 0.001$ (Chi Square test and *t*-test for age).

## 3. Results

The mean fear score of the FCV-19S with two additional questions was 22.6 (SD = 7.1), 20.5 (SD = 5.7) and 21.8 (SD = 6.4) among the medical, psychology and social work students, respectively ($F2_{,1982} = 18.665$; $p < 0.001$). The average fear value of all students was 21.9 (SD = 6.7). A Bonferroni post hoc test indicated that fear values differed between psychology and medical students ($p < 0.001$) and psychology and social work students ($p = 0.031$). The one-way ANOVA showed no significant differences in COVID-19 fear values based on year of study ($F3_{,1975} = 2.115$; $p = 0.096$). Regarding gender and religiosity, fear was lower among male than female ($t_{1976} = 7.900$; $p < 0.001$) students and secular than non-secular students ($t_{1981} = 8.859$; $p < 0.001$).

Table 2 provides information about last month tobacco and alcohol use before the COVID-19 pandemic, current last month use, including binge drinking (usually defined as the consumption of five or more alcoholic beverages on one occasion (at the time of the survey), and current last month substance use responses indicating if increased use was because of COVID-19.

**Table 2.** Substance use by study area.

| | Study Area | | | |
|---|---|---|---|---|
| | **Medicine (*n* = 1287)** | **Psychology (*n* = 552)** | **Social work (*n* = 256)** | **Total (*n* = 2095)** |
| Cigarette smoking before COVID-19, % (*n*) | 19.7 (229) *** | 27.8 (130) *** | 32.7 (74) *** | 23.3 (433) |
| Alcohol use before COVID-19, % (*n*) | 39.7 (464) *** | 60.0 (284) *** | 54.8 (125) *** | 46.7 (687) |
| Last month cigarette smoking, % (*n*) | 20.5 (240) *** | 30.6 (144) *** | 34.2 (77) *** | 24.7 (461) |
| Last month alcohol use, % (*n*) | 38.8 (453) *** | 56.6 (269) *** | 55.1 (125) *** | 45.3 (847) |
| Last month binge drinking, % (*n*) | 4.6 (54) ** | 3.8 (18) ** | 9.6 (22) ** | 5.0 (94) |
| Last month cigarette smoking more than usual, % (*n*) | 5.9 (69) *** | 7.0 (33) *** | 13.3 (30) *** | 7.1 (132) |
| Last month alcohol use more than usual, % (*n*) | 5.7 (66) *** | 6.7 (32) *** | 13.3 (30) *** | 6.9 (128) |

** $p < 0.01$; *** $p < 0.001$ (Chi Square test).

Results evidence that COVID-19 fear is linked to depression, exhaustion, loneliness, nervousness and anger (see Tables 3 and 4). Students who reported COVID-19 associated with their psycho-emotional well-being had higher fear values (Figure 1).

**Table 3.** COVID-19 fear impact on university medical, psychology and social work students' psycho-emotional well-being.

| | Study Area | | | |
|---|---|---|---|---|
| | **Medicine (*n* = 1287)** | **Psychology (*n* = 552)** | **Social work (*n* = 256)** | **Total (*n* = 2095)** |
| Depressed, % (*n*) | 26.4 (302) | 24.1 (110) | 30.2 (65) | 26.2 (477) |
| Exhausted, % (*n*) | 18.1 (204) | 21.7 (100) | 19.9 (43) | 19.2 (347) |
| Lonely, % (*n*) | 25.1 (285) | 25.1 (114) | 31.0 (66) | 25.8 (465) |
| Nervous, % (*n*) | 32.7 (376) | 30.5 (142) | 36.8 (81) | 32.6 (599) |
| Angry, % (*n*) | 22.7 (259) | 23.6 (108) | 28.8 (61) | 23.7 (428) |
| Any conditions, % (*n*) | 49.4 (551) | 51.0 (228) | 52.2 (108) | 50.1 (887) |

**Table 4.** COVID-19 fear impact on university students based on gender and religiosity.

| | Gender | | Religiosity | |
|---|---|---|---|---|
| | **Male (*n* = 513)** | **Female (*n* = 1575)** | **Secular (*n* = 725)** | **Non-secular (*n* = 1366)** |
| Depressed, % (*n*) | 17.8 (80) *** | 29.1 (396) *** | 25.9 (160) | 26.4 (316) |
| Exhausted, % (*n*) | 15.4 (69) * | 20.5 (277) * | 20.0 (123) | 18.9 (224) |
| Lonely, % (*n*) | 22.0 (99) * | 27.1 (365) * | 29.0 (180) * | 24.1 (284) * |
| Nervous, % (*n*) | 22.2 (100) *** | 36.0 (497) *** | 32.0 (200) | 33.1 (399) |
| Angry, % (*n*) | 21.0 (96) | 24.6 (332) | 25.4 (158) | 22.7 (269) |
| Any conditions, % (*n*) | 38.8 (171) *** | 53.9 (714) *** | 49.9 (301) | 50.3 (585) |

* $p < 0.05$; *** $p < 0.001$ (Chi Square test).

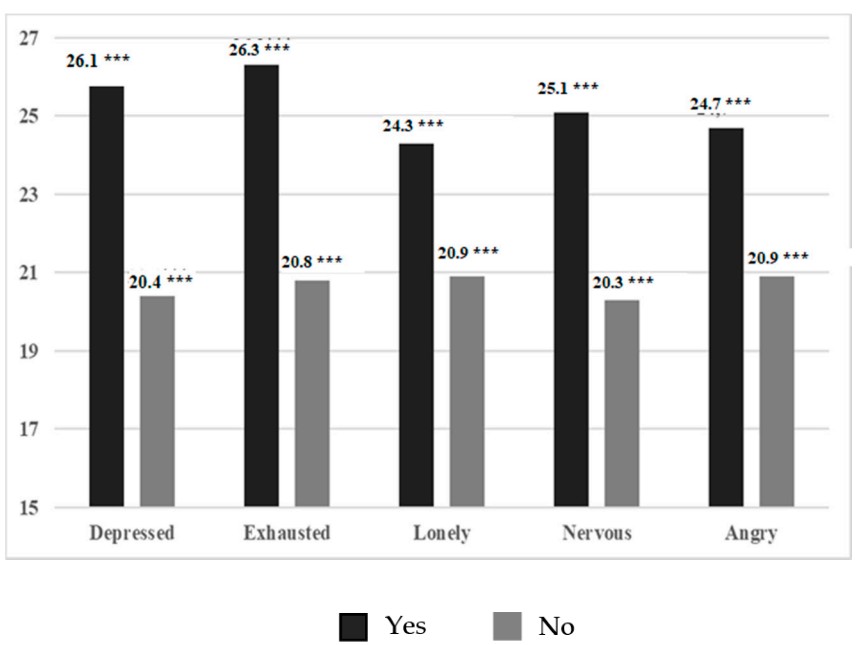

**Figure 1.** COVID-19 fear impact by psycho-emotional well-being among university students. (*** $p < 0.001$ (*t*-test)).

The two-way ANOVA showed significant change in fear values linked with last month binge drinking and students' psycho-emotional well-being associated with COVID-19: $F1_{,1704} = 9.487$; $p = 0.002$ (Figure 2). For students who reported deterioration of their psycho-emotional status, binge drinking cessation was associated with decreased fear.

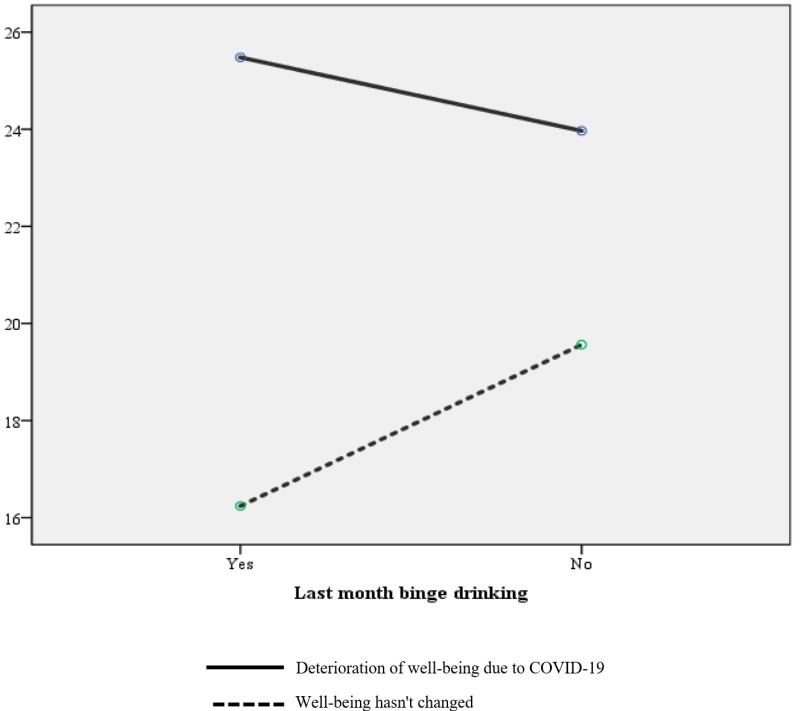

**Figure 2.** Fear of COVID-19 by last month binge drinking and psycho-emotional well-being due to COVID-19.

Regarding resilience values, the one-way ANOVA evidenced no significant difference among students based on study areas of medicine, psychology or social work ($F2_{,1854} = 1.862$;

$p = 0.156$). Regarding gender, males reported more resilience ($t_{1850} = 9.227$; $p < 0.001$), and no differences were found for resilience based on student secular and non-secular status ($t_{1852} = 1.730$; $p = 0.084$). A negative correlation was found between COVID-19 fear and resilience ($r = -0.311$; $p < 0.001$) meaning students with higher resilience tended to report lower COVID-19 fear. Students who reported more than usual substance use, binge drinking and psycho-emotional problems had lower resilience values (see Figure 3).

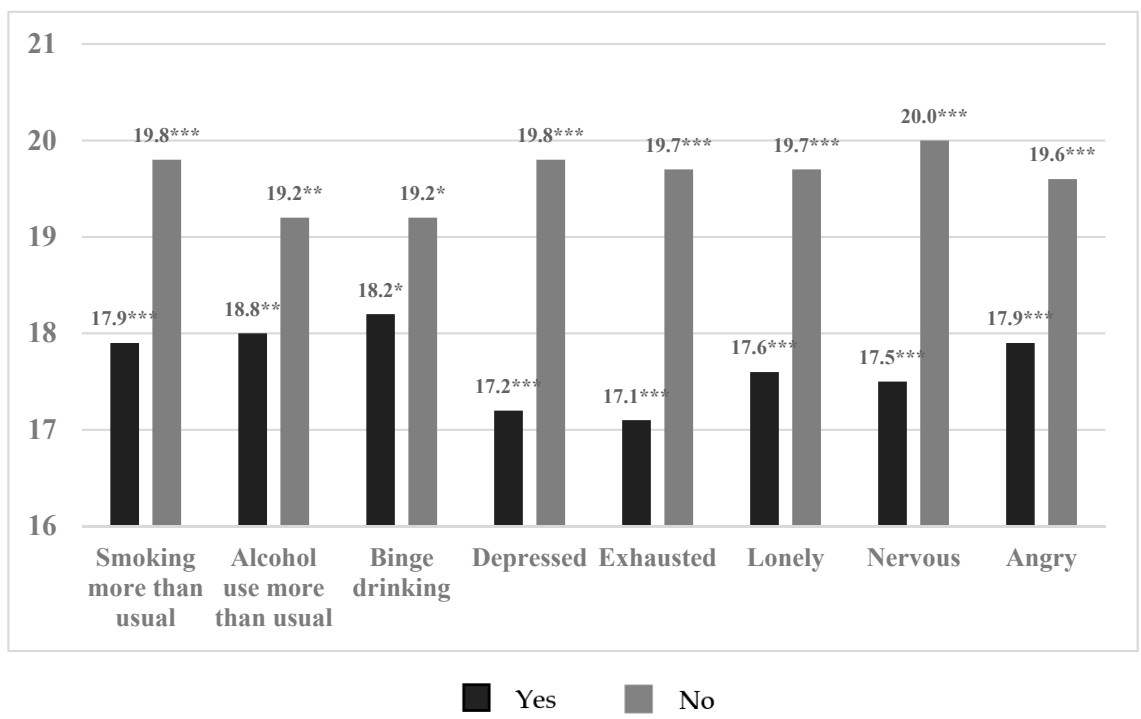

**Figure 3.** Resilience impact by substance use and psycho-emotional well-being among university students. (* $p < 0.05$; ** $p < 0.01$; *** $p < 0.001$ (*t*-test)).

## 4. Discussion

The present study, to the best of our knowledge, is the first investigation of COVID-19 fear among Russian university students from three helping professions (medicine, psychology and social work).

The hypothesis that COVID-19 fear, resilience, mental health and substance use differ among Russian students depending on study area was confirmed for fear and substance use only. Tobacco and alcohol use, during the October/November peak time of COVID-19 infection in Russia, was more prevalent among psychology and social work students than those from medicine. Depression, exhaustion, loneliness, nervousness and anger was more prevalent among females who reported a higher level of COVID-19-related fear and less resilience than males.

Results show a higher level of fear in women compared to men. This is consistent with other studies that also report higher levels of fear of COVID-19 in women and offer a number of explanations to better understand this global trend [14–16].

Non-secular (religious) students evidenced a higher level of fear. However, findings reflect a similar resilience level regardless of religious status. This outcome is contrary to the belief that religiosity is a protective factor that helps a person overcome difficult life circumstances [17,18]. In Russia, since the end of the 1980's, many people in Russia have affiliated with a religion. However, most people who identify with a religion tend to have a low level of practicing religious tenets, such as regular attendance at a place of worship [19]. This said, even though results showed a high religious level (65%) among "helping profession" students, it appeared to have little impact on resilience.

Based on study findings, the FCV-19S and the BRS demonstrated reliability as instruments for further research of COVID-19-related psycho-emotional problems, substance use and fears. Such scales may be useful for the study of other emergency/disaster conditions as well. Furthermore, future research should take into account the high prevalence of female service providers and students in the helping professions, such as medicine, nursing, psychology and social work, who may need tailored support and intervention [20,21].

## 5. Limitations

The present study has limitations. First, an online survey makes it difficult to generate random samples since the respondents were only those who use the Internet. Such sampling weakens generalizability of the study findings and does not provide survey participants the opportunity to ask clarifying questions. Second, further investigation over time and across locations is needed to better understand the association of COVID-19 fear, mental health, substance use and resilience among students, addressing pandemic and other disaster conditions.

## 6. Conclusions

The present study findings evidence the impact of COVID-19 on Russian medical, psychology and social work students. Further research is needed about the impact of social distancing and isolation that may be associated with student health and well-being, substance use and resilience. Furthermore, attention needs be given to students who assist medical, mental health and social work personnel on the front line of service provision in terms of any mental health conditions that they may experience from service to address the lingering pandemic.

**Author Contributions:** Conceptualization, V.K., V.G., A.R. and R.I.; methodology, V.K., A.R. and R.I.; validation, V.G.; formal analysis, A.R.; research, V.K., V.G. and A.R.; resources, V.K., V.G. and R.I.; data curation, V.G., R.Z., E.K. and I.O.; writing—original draft preparation, A.R.; writing—review and editing, V.K. and R.I.; visualization, R.Z., E.K. and I.O.; supervision, R.I. and A.R.; project administration and editing, R.I. All authors have read and agreed to the published version of the manuscript.

**Funding:** No external grant support specific to this research study was received.

**Institutional Review Board Statement:** Ethical review and approval were waived for this study due to the anonymous data collection and reporting procedures used. Further details are available from the corresponding author regarding the approvals received from the participating universities involved.

**Informed Consent Statement:** Informed consent was obtained from all subjects involved in the study.

**Data Availability Statement:** The data presented in this study are available on request from the corresponding author. The data are not publicly available due to local restrictions.

**Acknowledgments:** We acknowledge Toby and Mort Mower for their generous support of the Ben Gurion University of the Negev-Regional Alcohol and Drug Abuse Research (RADAR) Center.

**Conflicts of Interest:** The authors declare no conflict of interest.

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
