# Peer review of "COVID 19 Fear Impact among Russian “Helping Profession” Students"

_psych, doi:10.3390/psych3020014_

Round 1
Reviewer 1 Report
This is an important area of research. There are, however, areas of information and analysis needed as noted in the points below.
Information to be added/clarified:
-More information on the advertising process of the study to students. Was it distributed to all enrolled students? Selected students? Course-based distribution?
-More information is needed on the university ethics approval process. Was the completion of the questionnaire completely voluntary? If a student needed emotional support after completing the questionnaire was this available and indicated on the information letter?
-Please provide the logic behind the research design and why, for example was there an inclusion of the specific variable of religion?
-Some cultural and other information is needed as to the formation of the hypotheses when introduced, and also to be developed in the discussion section.
-Please provide the religious (secular/non-secular) question(s) included. Was an established instrument used? Why or why not?
-Were other demographic questions included? If yes, please include in the analysis.
-Please provide the extra questions added to the FCV-19S scale. This information needs to be included in the analysis process as well.
-Information is needed on the national rate of response by gender and size of program for the three professional institutions. How do the numbers correspond to the national averages? Or at least to the enrollment figures of the institutions involved. How many institutions were involved?
-In discussion some analysis by gender is needed. Why the difference? Cultural and/or other possible explanations? When sufficient information on religion is provided, this also needs to be discussed related to cultural and/or other possible explanations. Also, important to set this study in relation to both other similar Covid-related studies and to prior research in cultural context.
-Depending on the new information to be provided, perhaps a few more points to be added to the limitations of the study.
-No funding indicated but support in the acknowledgement implies funding of some nature. Please explain.
-Please check subject-verb agreement . Some errors found.
Author Response
Reviewer 1:
This is an important area of research. There are, however, areas of information and analysis needed as noted in the points below.
Information to be added/clarified:
-More information on the advertising process of the study to students. Was it distributed to all enrolled students? Selected students? Course-based distribution?
Since universities went into remote teaching mode, an online survey was conducted. Medical students, psychologists and social workers were informed about the survey by curators of the groups and through social networks. Sampling was not random. All interested students of “helping professions” (medicine, psychology, social work) could take part in the survey.
-More information is needed on the university ethics approval process. Was the completion of the questionnaire completely voluntary? If a student needed emotional support after completing the questionnaire was this available and indicated on the information letter?
Survey participation was anonymous and voluntary. Informational letter contained confirmation of survey anonymity and clarified the possibility of interruption at any stage. Names and email addresses of those responsible for conducting the study were also provided, who could be contacted if necessary. For additional explanations and other forms of support, students could contact their groups curators.
-Please provide the logic behind the research design and why, for example was there an inclusion of the specific variable of religion?
Despite the fact that Russia is a secular state, the country's leadership does not prevent and often encourages the entry of religious institutions into university space. In particular, the number of universities that have Orthodox churches on their territory is growing. Religion in Russia is viewed as a potential for moral and spiritual development of an individual. As our previous research shows (for example, https://doi.org/10.1016/j.ctim.2020.102546), students' religiosity is a factor influencing various aspects of their life and learning. Therefore, this variable is included in all studies that we have conducted in recent years.
-Some cultural and other information is needed as to the formation of the hypotheses when introduced, and also to be developed in the discussion section.
-Please provide the religious (secular/non-secular) question(s) included. Was an established instrument used? Why or why not?
We did not assess the degree of participants religiosity and therefore did not use any measuring scales for these purposes. In this study, we used a phenomenological approach and limited ourselves to dividing the participants into two categories depending on their answer to the question whether they are secular or religious.
-Were other demographic questions included? If yes, please include in the analysis.
see the text of the article (table 1)
-Please provide the extra questions added to the FCV-19S scale. This information needs to be included in the analysis process as well.
see the text of the article
-Information is needed on the national rate of response by gender and size of program for the three professional institutions. How do the numbers correspond to the national averages? Or at least to the enrollment figures of the institutions involved. How many institutions were involved?
Study involved 9 universities from different regions of Russia which train doctors, psychologists and social workers, including three medical universities. Response rates can only be estimated for medical universities using information on total number of students and number of students who took part in the survey. Response rate ranges from 2.5% for the Far Eastern State Medical University to 8.0% for Smolensk State Medical University. In other cases, in order to determine the level of response, inaccessible information is required on the number of students enrolled in individual faculties or individual departments of psychology and social work.
-In discussion some analysis by gender is needed. Why the difference? Cultural and/or other possible explanations? When sufficient information on religion is provided, this also needs to be discussed related to cultural and/or other possible explanations. Also, important to set this study in relation to both other similar Covid-related studies and to prior research in cultural context.
see the text of the article
-Depending on the new information to be provided, perhaps a few more points to be added to the limitations of the study.
-No funding indicated but support in the acknowledgement implies funding of some nature. Please explain.
-Please check subject-verb agreement. Some errors found.
Reviewer 2 Report
The topic is interesting and topical, but reading the article has given me some doubts.
The need for the study is not justified, the concepts analysed in the introduction are not explained and clarified. There is a lack of consistency in the theoretical framework.
The method does not indicate whether the instruments used have dimensions and/or how many dimensions they have. It is said that they are answered with a 5-point Likert-type scale, but it is not indicated what these points are.
It is stated that "the influence of COVID-19 on the psychoemotional well-being of students (i.e. depression, exhaustion, loneliness, nervousness and anger) was examined, but I have not found in the text, how it was done, which instrument was used to measure, which scale was used, validity and reliability of the instrument used.
For all the above reasons, I have to propose the non-acceptance of the article.
Author Response
Thank you for the comment.
Reviewer 3 Report
The study contributes to the literature on Covid-19's fear of students' mental health. The scientific quality of the study is obvious. One of the novelties is regarding the evaluation of this aspect in the case of helping professions. Interesting result regarding religious experience. However, the Method paragraph does not clearly specify which instrument was used to measure psycho-emotional well-being (both FCV -19 S and BRS have a single factor) and the frequency of alcohol and cigarette consumption. It is good if the authors bring clarifications in this regard by introducing a short sentence.
Author Response
Reviewer 3:
The study contributes to the literature on Covid-19's fear of students' mental health. The scientific quality of the study is obvious. One of the novelties is regarding the evaluation of this aspect in the case of helping professions. Interesting result regarding religious experience. However, the Method paragraph does not clearly specify which instrument was used to measure psycho-emotional well-being (both FCV -19 S and BRS have a single factor) and the frequency of alcohol and cigarette consumption. It is good if the authors bring clarifications in this regard by introducing a short sentence.
(see the text of the article)
Round 2
Reviewer 1 Report
Reviewer 1 FOR REVISED MANUSCRIPT:
The changes and additions add to a greatly improved text. However, there is still some work to be done. I have removed the points already addressed. The following points need attention.
Information to be added/clarified:
ADDITIONAL POINT IN REVISED VERSION:
The additional information on the translation process is important. However, there is still no information on any testing of these measures for comprehension and validity in the context's population. Were they tested at all in any student groups at least?
If yes, important to add to text. If not, important to indicate this AND add to limitation section.
POINTS NOT FULLY ADDRESSED:
-More information on the advertising process of the study to students. Was it distributed to all enrolled students? Selected students? Course-based distribution?
Since universities went into remote teaching mode, an online survey was conducted. Medical students, psychologists and social workers were informed about the survey by curators of the groups and through social networks. Sampling was not random. All
part in the survey.
THIS IS IMPORTANT INFORMATION TO BE INCLUDED.
-More information is needed on the university ethics approval process. Was the completion of the questionnaire completely voluntary? If a student needed emotional support after completing the questionnaire was this available and indicated on the information letter?
Survey participation was anonymous and voluntary. Informational letter contained confirmation of survey anonymity and clarified the possibility of interruption at any stage. Names and email addresses of those responsible for conducting the study were also provided, who could be contacted if necessary. For additional explanations and other forms of support, students could contact their groups curators.
THIS IS IMPORTANT INFORMATION TO BE INCLUDED. PLEASE ALSO EXPLAIN IN THE PAPER THE ROLE OF A CURATOR.
-Please provide the logic behind the research design and why, for example was there an inclusion of the specific variable of religion?
Despite the fact that Russia is a secular state, the country's leadership does not prevent and often encourages the entry of religious institutions into university space. In particular, the number of universities that have Orthodox churches on their territory is growing. Religion in Russia is viewed as a potential for moral and spiritual development of an individual. As our previous research shows (for example, https://doi.org/10.1016/j.ctim.2020.102546), students' religiosity is a factor influencing various aspects of their life and learning. Therefore, this variable is included in all studies that we have conducted in recent years.
THIS NEEDS TO BE INCLUDED IN SHORTEMED FORM UNDER METHODS AND WITH A FORMULATION OF THE SPECIFIC QUESTION ASKED. WHEN EXAMING THE IMPORTANT INFORMATION YOU ADDED TO THE DISCUSSION AS TO THE DISTINCTION BETWEEN CATEGORY AND PRACTICE, WHY WAS A PRACTICE QUESTION NOT ASKED AS WELL? THESE AREAS ARE NOW RECOMMENDED IN RESEARCH ON THE TOPIC OF RELIGION. THIS SHOULD BE LINCLUDED AS A LIMITATION OF EXPLORING THIS VARIABLE.
-Please provide the religious (secular/non-secular) question(s) included. Was an established instrument used? Why or why not?
We did not assess the degree of participants religiosity and therefore did not use any measuring scales for these purposes. In this study, we used a phenomenological approach and limited ourselves to dividing the participants into two categories depending on their answer to the question whether they are secular or religious. A phenomenological approach can be used to add a question as to practice.
-Information is needed on the national rate of response by gender and size of program for the three professional institutions. How do the numbers correspond to the national averages? Or at least to the enrollment figures of the institutions involved. How many institutions were involved?
Study involved 9 universities from different regions of Russia which train doctors, psychologists and social workers, including three medical universities. Response rates can only be estimated for medical universities using information on total number of students and number of students who took part in the survey. Response rate ranges from 2.5% for the Far Eastern State Medical University to 8.0% for Smolensk State Medical University. In other cases, in order to determine the level of response, inaccessible information is required on the number of students enrolled in individual faculties or individual departments of psychology and social work.
NEEDS TO BE INCLUDED IN THE ARTICLE. THE SPECIFIC NAME OF THE MEDICAL UNIVERSITIES MAY BE DELETED.
-Please check subject-verb agreement. Some errors found.
Several remaining errors! Verb-subject agreement, repetition of words, often missing words such as ‘the’ before measures.
Author Response
We express our sincere gratitude to the reviewer for a detailed, deep and very useful analysis for our work. Referring to the question about whether the tools used were checked / validated on a Russian-speaking sample. Our answer is yes, of course. We did a test (primarily for the Fear of COVID-19 Scale) on a sample of 850 people and it is described in detail in the source (Reznik, A .; Gritsenko, V .; Konstantinov, V .; Khamenka, N .; Isralowitz, R. COVID-19 fear in Eastern Europe: validation of the fear of COVID-19 scale. Int. J. Ment. Health Addict. 2020. is reference number 8 in the list of references). The Brief Resilience Scale was also tested on a sample of Israeli university students.
Please see the attachment below also.

Reviewer 2 Report
All issues raised have been corrected
Author Response
We are appreciative of this comment.
Round 3
Reviewer 1 Report
The text is fine now. Please correct for consistency in reference formatting.